# IgA2 Antibodies against SARS-CoV-2 Correlate with NET Formation and Fatal Outcome in Severely Diseased COVID-19 Patients

**DOI:** 10.3390/cells9122676

**Published:** 2020-12-12

**Authors:** Léonie A. N. Staats, Hella Pfeiffer, Jasmin Knopf, Aylin Lindemann, Julia Fürst, Andreas E. Kremer, Holger Hackstein, Markus F. Neurath, Luis E. Muñoz, Susanne Achenbach, Moritz Leppkes, Martin Herrmann, Georg Schett, Ulrike Steffen

**Affiliations:** 1Department of Internal Medicine 1, Universitätsklinikum Erlangen, Friedrich-Alexander-Universität Erlangen-Nürnberg (FAU), 91054 Erlangen, Germany; Leonie.Staats@uk-erlangen.de (L.A.N.S.); Aylin.Lindemann@uk-erlangen.de (A.L.); Julia.Fuerst@uk-erlangen.de (J.F.); Andreas.Kremer@uk-erlangen.de (A.E.K.); Markus.Neurath@uk-erlangen.de (M.F.N.); Moritz.Leppkes@uk-erlangen.de (M.L.); 2Deutsches Zentrum Immuntherapie (DZI), 91054 Erlangen, Germany; Jasmin.Knopf@uk-erlangen.de (J.K.); Luis.Munoz@uk-erlangen.de (L.E.M.); Martin.Herrmann@uk-erlangen.de (M.H.); Georg.Schett@uk-erlangen.de (G.S.); 3Department of Transfusion Medicine and Haemostaseology, Universitätsklinikum Erlangen, Friedrich-Alexander-Universität Erlangen-Nürnberg (FAU), 91054 Erlangen, Germany; Hella.Pfeiffer@uk-erlangen.de (H.P.); Holger.Hackstein@uk-erlangen.de (H.H.); Susanne.Achenbach@uk-erlangen.de (S.A.); 4Department of Internal Medicine 3, Universitätsklinikum Erlangen, Friedrich-Alexander-Universität Erlangen-Nürnberg (FAU), 91054 Erlangen, Germany

**Keywords:** IgA, SARS-CoV-2, COVID-19, inflammation, neutrophil extracellular trap (NET)

## Abstract

Infection with severe acute respiratory syndrome coronavirus 2 (SARS-CoV-2) leads to an adaptive immune response in the host and the formation of anti-SARS-CoV-2 specific antibodies. While IgG responses against SARS-CoV-2 have been characterized quite well, less is known about IgA. IgA2 activates immune cells and induces inflammation and neutrophil extracellular trap (NET) formation which may contribute to organ injury and fatal outcome in SARS-CoV-2-infected patients. SARS-CoV-2 spike protein specific antibody levels were measured in plasma samples of 15 noninfected controls and 82 SARS-CoV-2-infected patients with no or mild symptoms, moderate symptoms (hospitalization) or severe disease (intensive care unit, ICU). Antibody levels were compared to levels of C-reactive protein (CRP) and circulating extracellular DNA (ecDNA) as markers for general inflammation and NET formation, respectively. While levels of SARS-CoV-2-specific IgG were similar in all patient groups, IgA2 antibodies were restricted to severe disease and showed the strongest discrimination between nonfatal and fatal outcome in patients with severe SARS-CoV-2 infection. While anti-SARS-CoV-2 IgG and IgA2 levels correlated with CRP levels in severely diseased patients, only anti-SARS-CoV-2 IgA2 correlated with ecDNA. These data suggest that the formation of anti-SARS-CoV-2 IgA2 during SARS-CoV-2 infection is a marker for more severe disease related to NET formation and poor outcome.

## 1. Introduction

In 2020, severe acute respiratory syndrome coronavirus 2 (SARS-CoV-2) has induced a pandemic disease (coronavirus disease of 2019 (COVID-19)) with more than 37 million people infected and over 1 million deaths worldwide (status from 11 October 2020) [1]. Although in many cases infected people do not suffer from severe disease and might even remain asymptomatic, a significant portion of SARS-CoV-2-infected subjects have to be treated in a hospital, with one-third of them requiring mechanical ventilation in an intensive care unit (ICU) [2,3,4].

The role of the immune system and especially of antibodies against SARS-CoV-2 is controversially discussed. It is assumed that in the early phase of the disease, antibodies help to clear the virus and thus contribute to controlling the infection. Indeed, increased antibody levels at early stages of COVID-19 were found to correlate with a decreased viral load and better survival of the patients [5]. On the other hand, in hospitalized COVID-19 patients, high antibody titers are reported to be associated with worse outcome [6,7]. SARS-CoV-2-specific antibodies likely form immune complexes together with their viral antigens. Immune complexes activate the complement system and innate immune cells, such as macrophages or neutrophils. The formation of SARS-CoV-2-containing immune complexes may thus enhance local or systemic inflammation. There is evidence that SARS-CoV-2 infection can not only cause alveolar epithelial cell damage due to the viral activity but also lead to a cytokine-storm-like hyperinflammation that provokes further injury [8,9].

In addition, neutrophil activation and the formation of neutrophil extracellular traps (NETs) have been described as major risk factors for mortality in COVID-19 patients [10]. Sera from COVID-19 patients show elevated levels of markers for NET formation, such as circulating extracellular DNA (ecDNA), neutrophil elastase (NE) activity or myeloperoxidase-DNA (MPO-DNA), and these levels correlate with disease severity [11,12,13]. NETs have been found in the microvasculature of lung, kidney and heart biopsies of deceased COVID-19 patients and are thought to contribute to immunothrombosis-mediated damage of these organs [12,13,14]. Sera from COVID-19 patients have been shown to trigger NET formation [11,15]. Although some groups described a direct positive effect of the SARS-CoV-2 virus on NET formation [16,17], it seems likely that immune complexes formed by SARS-CoV-2 and SARS-CoV-2-specific antibodies in the serum of COVID-19 patients additionally trigger NET formation.

While macrophages are effectively activated by IgG immune complexes, neutrophils mainly respond to immune complexes containing IgA [18,19,20]. In human sera, IgA represents the second most prevalent immunoglobulin class and is associated with several autoimmune diseases [21]. Humans possess two IgA subclasses, IgA1 and IgA2. We have recently shown that IgA2 especially elicits proinflammatory effector functions in neutrophils [22]. Hence, we were interested to analyze SARS-CoV-2-specific IgA2 levels and to relate them to the clinical course of COVID-19, the inflammatory state and markers for NET formation in SARS-CoV-2-infected subjects.

## 2. Materials and Methods

### 2.1. Patients and Healthy Controls

Fifteen healthy controls and 82 SARS-CoV-2-infected patients diagnosed by positive RNA tests from oral, nasal and tracheal swabs were recruited in the University Hospital Center of Erlangen (Bavaria, Germany). All experiments were performed in accordance with the institutional guidelines and the agreement of the local ethics committee (permit #277_17B, #125_13B and #174_20B). SARS-CoV2-infected patients were grouped by disease severity: subjects with no/mild disease symptoms who were convalescent at the time of blood draw (N = 34), subjects with moderate disease requiring hospitalization (N = 31) and subjects with severe disease requiring intensive care (N = 17). The decision of whether a patient had to be hospitalized was made at the discretion of the treating physician. Indications for hospitalization were, amongst others, continuous fever, dyspnea, the presence of serious comorbidities or a poor overall health condition. All patients who were transferred to an intensive care unit had to be ventilated. Patient characteristics are described in Table 1.

### 2.2. Measurement of Antibody Levels

Antibody levels were measured in plasma samples from the above-described groups with ELISA using SARS-CoV-2 spike S1+S2 protein (#40589-V08B1; Sino Biologicals, Wayne, PA, USA) as a capture antigen. Plates were coated overnight at 4 °C with 2 µg/mL of the capture antigen in a 0.05 M carbonate–bicarbonate buffer and blocked with 1% bovine serum albumin (#0163.4; Carl Roth GmbH, Karlsruhe, Germany) in phosphate-buffered saline (#14190-094; Thermo Scientific, Waltham, MA, USA). Plasma samples were diluted 1:2400 for IgG and 1:200 for IgA1 and IgA2 detection. As detection antibodies we used horseradish peroxidase coupled goat-anti-IgG (#2040-05), mouse-anti-IgA1 (#9130-05) or mouse-anti-IgA2 (#9140-05; all Southern Biotech, Birmingham, AL, USA). Plates were developed with 3,3′,5,5′-tetramethylbenzidine (TMB) (#34021; Thermo Scientific, Waltham, MA, USA). Optical density (OD) was measured at 450 nm and corrected against OD values at 650 nm.

### 2.3. Measurement of C-Reactive Protein (CRP), Extracellular DNA (ecDNA) and Neutrophil Elastase (NE) Activity

C-reactive protein (CRP) was measured by nephelometry. The amount of ecDNA in plasma was quantified with the Quant-iT PicoGreen dsDNA Assay-Kit (#P11496; Thermo Scientific, Waltham, MA, USA) according to the manufacturer’s instructions. Briefly, samples were diluted 1:20 in a TE buffer and Quant-iT PicoGreen dsDNA reagent was added in a 1:1 ratio followed by incubation for 5 min at room temperature in the dark. Fluorescence was measured in an Infinite 200 PRO plate reader (Tecan, Männedorf, Switzerland) with excitation at 485 nm and emission at 535 nm. The concentration of the ecDNA was calculated with the standard provided by the kit. NE activity was assessed in plasma with the fluorogenic substrate MeOSuc-AAPV-AMC (sc-201163; Santa Cruz Biotechnology, Dallas, TX, USA) to a final concentration of 100 μM at 37 °C. Fluorescent readings at 37 °C were collected on a TECAN Infinite 200 Pro using the filter set ex. 360 nm, em. 465 nm for 24 h in a 20 min interval.

### 2.4. Statistical Analysis

Statistical analysis was performed with Prism 8.3 (GraphPad Software, San Diego, CA, USA). For comparison of two groups, a two-sided Mann–Whitney U test was used. Statistics for three or more groups were calculated with the Kruskal–Wallis test followed by Dunn’s multiple comparison test for all groups vs. the control group. Correlations were investigated with Spearman’s correlation coefficient. *p* < 0.05 was considered significant. Data are presented as scatter plots with mean ± standard error of mean (s.e.m.) or scatter plots. All analyses were performed in a blinded manner.

## 3. Results

### 3.1. Comparison of SARS-CoV-2 Antibody Levels in COVID-19 Patients with Different Disease Severity and Outcome

Eighty-two SARS-CoV-2-infected individuals and 15 healthy controls were analyzed in this study. Demographic characteristics are depicted in Table 1.

In general, patients with moderate and severe courses of SARS-CoV-2 were older and had more comorbidities as compared to those with mild disease.

When analyzing antibody levels directed against the S1+S2 spike protein of SARS-CoV-2 in the plasma of these subjects, we found that IgG levels were profoundly increased in all patient groups compared to healthy controls (Figure 1a). In contrast, anti-SARS-CoV-2 IgA responses clustered in subjects with moderate and severe disease. Interestingly, IgA2 responses were virtually confined to severely diseased patients requiring treatment at an ICU. We next compared levels of SARS-CoV-2-specific antibodies between severely diseased patients who recovered and patients who died from infection. While the levels of all antibodies against SARS-CoV-2 were higher in patients with fatal outcome, these differences were strongest for IgA2 antibodies (Figure 1b).

As patients with moderate and severe disease were older, we tested if SARS-CoV-2-specific antibody levels correlated with age in these groups. In patients with severe disease, but not moderate disease, anti-SARS-CoV-2 IgA1 and IgA2 were slightly increased in aged patients (Appendix A).

### 3.2. Relation of SARS-CoV-2-Specific Antibody Levels to Systemic Inflammation and NET Formation

To investigate if SARS-CoV-2-specific antibodies may contribute to fatal disease outcome in severely diseased patients, we first compared their antibody levels with the systemic inflammation marker CRP. We excluded all patients that had developed an additional bacterial infection from this analysis as this might have affected the CRP values. In severely diseased patients, serum levels of anti-SARS-CoV-2 IgG or IgA2 correlated with CRP levels (Figure 2a), suggesting the inflammation was in general linked to a more robust antiviral immune response in these patients. Of note, we did not observe any correlation between SARS-CoV-2-specific IgG, IgA1 or IgA2 levels and CRP values in patients with moderate disease activity (Figure 2b).

Recently, it has been described that the formation of NETs induces vascular occlusion and subsequent organ damage [12,13]. Patients with moderate or severe disease showed elevated amounts of ecDNA and NE activity (Appendix A), indicating an increased NET formation rate in these patients. As IgA2 is a potent inducer of NET formation [22], we hypothesized that SARS-CoV-2-specific IgA2 might, in addition to the induction of hyperinflammation, contribute to organ damage by increasing NET formation. To test this, we compared the levels of SARS-CoV-2-specific IgA2 in severely diseased patients to the amount of ecDNA as a surrogate marker for NET formation. To avoid the results being biased by NET formation as a response to invading pathogens, we excluded all patients that had developed an additional bacterial infection from this analysis. Interestingly, SARS-CoV-2-specific IgA2, but not IgG or IgA1 levels, correlated with the amount of ecDNA (Figure 2c), suggesting that IgA2 specifically contributes to NET formation in SARS-CoV-2 infection.

## 4. Discussion

The role of SARS-CoV-2-specific antibodies in the pathogenesis of COVID-19 is still incompletely understood. Immune cell activation and antibody formation are required to eliminate the virus. On the other hand, there is evidence that the immune system might worsen the disease at later stages. COVID-19 is associated with the development of cytokine-storm-like hyperinflammation, and data indicate that elevated SARS-CoV-2-specific antibody levels are associated with a more severe disease [6,7].

In this study, we showed that SARS-CoV-2-specific IgA2 antibodies are enriched in more severe cases of COVID-19. In contrast, SARS-CoV-2-specific IgG was detected in a wide range of SARS-CoV-2-infected subjects, including those with mild, moderate and severe disease manifestations. Contrary to other studies [23,24], we did not see a significant difference between patients with mild or moderate and severe disease. The fact that OD values from a single dilution do not have a linear correlation with titers might contribute to the underestimation of the IgG response in patients with high levels. However, nearly all patients showed a robust IgG response against SARS-CoV-2, which was not the case for IgA2.

These results indicate that IgA and especially IgA2 antibodies against SARS-CoV-2 may be a better immunologic indicator for severe disease compared to IgG antibodies. This hypothesis is in accordance with studies describing higher SARS-CoV-2-specific, but also total, IgA levels in patients with severe COVID-19 compared to patients with moderate disease [25,26]. Antibody class switches from IgG to IgA1 or IgA2 are a frequent phenomenon [27]. The fact that we detected SARS-CoV-2-specific IgA2 mainly in severe COVID-19 cases suggests that the formation of IgA2 antibodies requires a stronger stimulus than the formation of IgG antibodies. Thus, the development of SARS-CoV-2-specific IgA2 may indicate a state of immune dysregulation associated with severe COVID-19. Overall, SARS-CoV-2-specific IgA seems to have a more dynamic course than IgG as it has also been shown that IgA specific for SARS-CoV-2, as well as SARS-CoV, disappears faster than the respective IgG [25,28].

In the literature, there is a discussion on the potential role of SARS-CoV-2-specific IgG in contributing to the hyperinflammation status of severely diseased COVID-19 patients [9,29,30]. IgG-based immune complexes are potent activators of macrophages and induce the release of inflammatory cytokines such as interleukin-6 [31], which promotes the hepatic release of CRP [32]. This is in accordance with our finding that SARS-CoV-2-specific IgG antibodies were related to elevated CRP levels in severely diseased patients. The correlation was not significant, which is most likely due to the relatively small cohort size. Interestingly, we did not see any correlation between SARS-CoV-2-specific IgG levels and CRP values in moderately diseased patients, although the anti-SARS-CoV-2 IgG levels were comparable to patients with severe disease. There was no difference regarding comorbidities between the two patient groups except an increased prevalence of chronic obstructive pulmonary disease (COPD) in patients with severe disease. A recent publication shows that SARS-CoV-2-specific IgG in the serum of severely diseased patients displays an aberrant Fc glycosylation profile, which results in increased macrophage activation [9]. Fc glycosylation is a key regulator of IgG effector functions [33], and alterations in the glycosylation profile towards a more inflammatory phenotype might explain the observed correlation between high SARS-CoV-2-specific antibody levels and systemic inflammation in severely diseased patients.

Our data showed that the IgA2 subclass especially is associated with severe coronavirus disease symptoms and fatal outcome. IgA2 is the primary proinflammatory effector component within the IgA class [22] and a potent inducer of NET formation [18,19,20,22]. Vascular occlusion caused by NET formation has been described as a major risk factor for COVID-19 patients and might be even more dangerous than hyperinflammation [10,11,12,13,14]. In concordance with published data [11,12], we found elevated levels of the NET surrogate markers ecDNA and NE activity. SARS-CoV-2-specific IgA2 levels correlated with increased amounts of ecDNA, suggesting that SARS-CoV-2-specific IgA2 may contribute to organ damage by the induction of NET formation and, consequently, vascular occlusion. It is conceivable that IgA2 immune complexes might functionally contribute to immunothrombosis in severe COVID-19. On the other hand, the increased presence of IgA1 and IgA2 might indicate a breached pulmonary barrier allowing the systemic spread of mucosa-targeted IgA. In addition, increased NET formation may be due to an increased viral load in severely diseased patients. SARS-CoV-2 has been shown to directly induce NET formation [16,17], and more research is needed to investigate whether antibodies are involved in this process.

Of note, in severely diseased patients, we found a correlation of SARS-CoV-2-specific IgA1 and IgA2 levels with age. This observation is in accordance with other studies showing that increased age is associated with stronger disease [24,34,35].

In this study, we did not investigate the presence of SARS-CoV-2-specific IgM. To our knowledge, IgM is not considered a potent trigger of NET formation but rather immunologically silent. Present studies suggest that SARS-CoV-2-specific IgM is less prevalent compared to IgG and that it is not associated with disease severity [7,36].

Taken together, our data indicate that SARS-CoV-2-specific antibodies are associated with severity of COVID-19. In this regard, especially elevated anti-SARS-CoV-2 IgA2 may serve as an indicator for severe disease related to NET formation and poor outcome.

## Figures and Tables

**Figure 1 cells-09-02676-f001:**
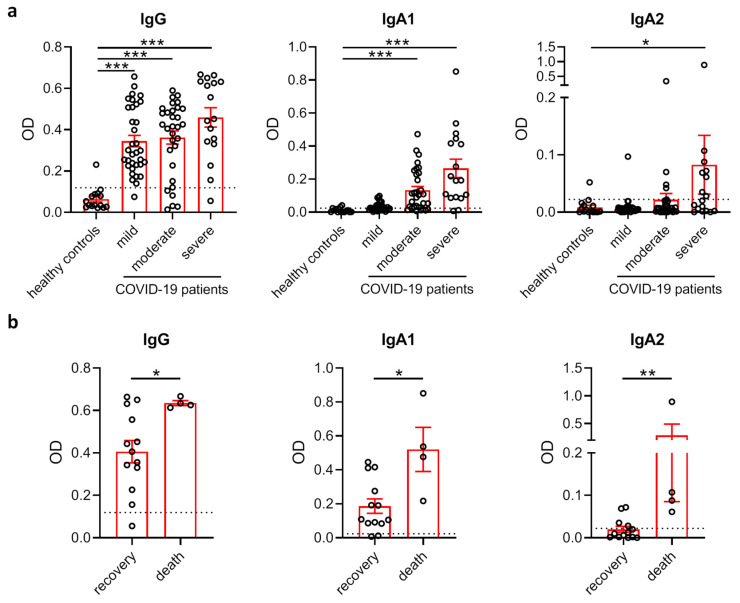
SARS-CoV-2-specific IgA2 levels correlate with disease severity. (**a**) Levels of IgG, IgA1 and IgA2 directed against S1+S2 protein of SARS-CoV-2 in the plasma of healthy control subjects (N = 15) as well as SARS-CoV-2-infected patients with no or mild disease symptoms (N = 34), moderate symptoms (N = 31) or severe symptoms requiring intensive care treatment (N = 17). Dotted lines represent mean + standard deviation (SD) of the healthy control group. (**b**) Levels of IgG, IgA1 and IgA2 directed against S1+S2 protein of SARS-CoV-2 in the plasma of COVID-19 patients with severe disease who recovered (N = 13) or died (N = 4). Dotted lines represent mean + SD of the healthy control group from (**a**). Significances were tested with the Kruskal–Wallis test followed by Dunn’s multiple comparison test for all groups vs. the control group (**a**) and a two-sided Mann–Whitney U test (**b**). * *p* < 0.05; ** *p* < 0.01; *** *p* < 0.001. Data are presented as scatter plots with mean ± s.e.m.

**Figure 2 cells-09-02676-f002:**
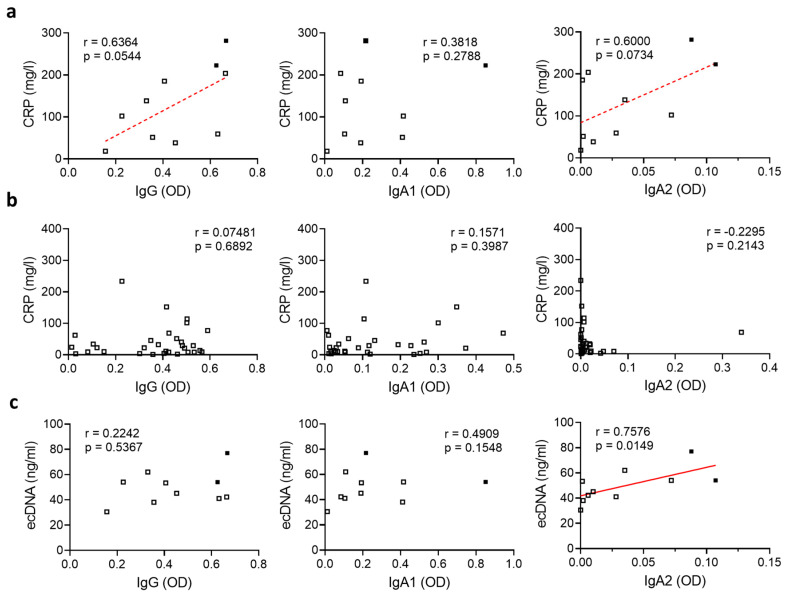
SARS-CoV-2-specific IgA2 levels correlate with ecDNA in severely diseased patients. (**a**,**b**) Correlation of levels of IgG, IgA1 and IgA2 directed against S1+S2 protein of SARS-CoV-2 with CRP in the plasma of SARS-CoV-2-infected patients with (**a**) severe disease (N = 10) or (**b**) moderate disease (N = 31). Filled squares represent patients who died. (**c**) Correlation of levels of IgG, IgA1 and IgA2 directed against S1+S2 protein of SARS-CoV-2 in the plasma of SARS-CoV-2-infected subjects with severe disease (N = 10) with the amount of ecDNA. Filled squares represent patients who died. Significance was tested with Spearman’s correlation coefficient. *p* < 0.05 was considered significant. Data are presented as scatter plots.

**Table 1 cells-09-02676-t001:** Characteristics of blood donors.

	Healthy Controls (N = 15)	Mild Disease (N = 34)	Moderate Disease * (N = 31)	Severe Disease ** (N = 17)
Age (ys): median (range)	34 (23–63)	39 (19–65)	68 (34–96)	68 (37–78)
Male, N (%)	5 (33%)	28 (82%)	17 (55%)	13 (76%)
Female, N (%)	10 (67%)	6 (18%)	14 (45%)	4 (24%)
CAD, N (%)	n.a.	0 (0%)	8 (26%)	5 (29%)
Hypertension, N (%)	n.a.	1 (3%)	18 (58%)	13 (76%)
Diabetes mellitus, N (%)	n.a.	2 (6%)	8 (26%)	6 (35%)
COPD, N (%)	n.a.	0 (0%)	1 (3%)	4 (24%)
Days in hospital: median (range)	0 (0–0)	0 (0–14)	8 (1–64)	31 (2–50)
Death, N (%) ***	0 (0%)	0 (0%)	0 (0%)	4 (24%)

* Hospitalized; ** requiring intensive care unit; *** due to SARS-CoV-2; CAD, coronary artery disease; COPD, chronic obstructive pulmonary disease; n.a., not analyzed.

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
