# Peer review of "IgA2 Antibodies against SARS-CoV-2 Correlate with NET Formation and Fatal Outcome in Severely Diseased COVID-19 Patients"

_cells, 2020, doi:10.3390/cells9122676_

Round 1

Reviewer 1 Report

  • Authors should contain on the "Materials and Methods" section the criteria/guidilines according to which the severity of the disease was determined. 
  • Authors should present on standalone diagrams the levels of NETs on plasma of those patients alongside healthy controls. Cell free DNA only is not acceptable as a protein marker is missing. Maybe, authors could use MPO/DNA complex elisa or CitH3 alongside with cell free DNA measurement.
  • Language in the "Discussion" section is a little bit too "strong". Data of the manuscript are not enough to support the phrase: " In this regard IgA2 seems to be more dangerous than IgG as it has the potential to induce NET formation."
  • The authors seem to ignore publications that concern NETs in COVID-19 such as:

Neutrophil extracellular traps in COVID-19. JCI Insight. 2020 Jun 4;5(11):e138999. doi: 10.1172/jci.insight.138999.

Complement and tissue factor–enriched neutrophil extracellular traps are key drivers in COVID-19 immunothrombosis. J Clin Invest. 2020 Nov 2;130(11):6151-6157. doi: 10.1172/JCI141374.

and others.

Author Response

Reviewer 1

Authors should contain on the "Materials and Methods" section the criteria/guidelines according to which the severity of the disease was determined.

We have added a description of the criteria for hospitalization and the transfer to an intensive care unit to the methods section (page 2; 2.1. Patients and healthy controls; line 81).

Authors should present on standalone diagrams the levels of NETs on plasma of those patients alongside healthy controls. Cell free DNA only is not acceptable as a protein marker is missing. Maybe, authors could use MPO/DNA complex elisa or CitH3 alongside with cell free DNA measurement.

Thank you for this suggestion. We have measured ecDNA in all groups and show the results in Supplementary Figure S2.  EcDNA is significantly increased in patients with moderate and severe disease. In addition, we have measured the activity of neutrophil elastase and compared it with the amounts of ecDNA. Both markers correlated with each other, indicating that ecDNA is a valid surrogate marker for NET formation. We have included this information to the manuscript (Figure S2). As for some patients no fresh aliquot was available anymore, the group sizes are slightly smaller compared to the previous measurements.

Language in the "Discussion" section is a little bit too "strong". Data of the manuscript are not enough to support the phrase: "In this regard IgA2 seems to be more dangerous than IgG as it has the potential to induce NET formation."

We have rewritten the last sentences of the discussion section (page 7, line 244) in order to soften the language.

The authors seem to ignore publications that concern NETs in COVID-19 such as:

Neutrophil extracellular traps in COVID-19. JCI Insight. 2020 Jun 4;5(11):e138999. doi: 10.1172/jci.insight.138999.

Complement and tissue factor–enriched neutrophil extracellular traps are key drivers in COVID-19 immunothrombosis. J Clin Invest. 2020 Nov 2;130(11):6151-6157. doi: 10.1172/JCI141374.

and others.

We have added a paragraph to the introduction (page 2, line 56) that describes the current knowledge about the role of neutrophils and NETs in COVID-19 in more detail. Furthermore, we have extended the respective paragraph in the discussion (page 7, line 226 and 230).

Reviewer 2 Report

In this manuscript, Staats et al. have studied antibody responses against SARS-CoV-2 in COVID-19 patients with different disease severity. They identified that disease severity correlates with IgA titers, particularly IgA2. In addition, the authors found that concentrations of extracellular DNA correlate with IgA2 titers.

The presented findings are very much in line with other published and preprint studies that show that anti-SARS-CoV-2 antibody titers correlate with COVID-19 severity. Yet, while others studies mainly focus on IgG, this study focuses on IgA, which in general is more relevant in the context of airway infections, including COVID-19. The authors show correlation of IgA with systemic inflammation and extracellular DNA, but do not provide functional data to prove that IgA directly contributes to inflammation or NET formation. Nevertheless, this provides important new clues that are likely to be relevant to explain disease severity in COVID-19.

I have no major concerns, but I do have a number of minor concerns that should be addressed:

  1. Throughout the manuscript, the antibody titers have been determined using OD. In general ODs do not have a linear correlation with the titers, but a so-called “S curve”, meaning that very high and very low titers show non-linear differences. This could underlie the lack of clearly increased IgG titers between mild-moderate-severe patients in Fig. 1a, something that is found by most other groups. In addition, the ODs for IgA2 are very, very low. Since IgA2 is the most important subclass in this manuscript, it would be good to know if these data are reliable. Therefore, for both IgG and IgA subclasses it would be good to include a standard curve to demonstrate the sensitivity and the reliable range of the ELISA.
  2. Since anti-SARS-CoV-2 titers change over time, when comparing samples it is important that these are taken at a similar stage of the disease (e.g. day after onset or hospitalization). However, nothing is mentioned about that in the manuscript.
  3. In Fig. 1a, it would be far more interesting to see whether there are statistical differences between mild-moderate-severe COVID-19 patients, instead of the comparison to healthy controls.
  4. The use a log scale for the IgA2 graph in Fig. 2a seems a bit inappropriate, since all others graphs are linear.
  5. In Fig. 2c the authors leave out patients with a bacterial superinfection, since it could be a confounding factor for NETosis. Yet, while a bacterial superinfection could also interfere with CRP levels, these patients are not left out in Fig. 2a and b. This requires an explanation.
  6. The authors merely show correlation, but do not provide evidence for a direct contribution of IgA2 to disease severity. Therefore, the final paragraph of the discussion should be toned down a bit. For example, change “may contribute to disease severity” into “are associated with disease severity”; and “IgA2 seems to be more dangerous” into “IgA2 may be more dangerous”.

Reviewer 3 Report

Staats and colleagues present an interesting study about the role of IgGA2 antibodies, NET formation and COVID19.

This report is well designed and performed with interesting findings.

Only minor comments to look into:

line 42: ..."infected subjects want".... Is there a better word than "want"?

Figure 2c: What does it look like for moderate disease and eDNA?

line 159: ..."system might worsen disease at later stage"... Add "the" between worsen and disease

line 174: remove "2" from SARS-CoV-2, as the reference 20 is only about SARS-CoV (published in 2004)

line 235: reference 9 not consistent, need to be change to "et al." after third author

Author Response

Staats and colleagues present an interesting study about the role of IgGA2 antibodies, NET formation and COVID19.

This report is well designed and performed with interesting findings.

Only minor comments to look into:

line 42: ..."infected subjects want".... Is there a better word than "want"?

We have rewritten this phrase.

Figure 2c: What does it look like for moderate disease and ecDNA?

We have included the values of ecDNA for all patients in Supplementary Figure S1. There was no difference between healthy controls and patients with no/mild symptoms. In contrast, patients with moderate or severe disease showed elevated levels of ecDNA. It seems that ecDNA is more abundant in plasma of patients with severe disease compared to patients with moderate disease. But this difference was not significant.

line 159: ..."system might worsen disease at later stage"... Add "the" between worsen and disease

Thank you for this remark. We have corrected the mistake.

line 174: remove "2" from SARS-CoV-2, as the reference 20 is only about SARS-CoV (published in 2004)

Thank you for this remark. We have corrected the respective part to “SARS-CoV-2 as well as SARS-CoV specific IgA”.

line 235: reference 9 not consistent, need to be change to "et al." after third author

We have corrected the appearance of reference 9.

Reviewer 4 Report

The topic of the manuscript is of high importance, it is still 2020. And it does contribute significant new data to the arising picture of the Sars-Cov-2 infection course. The novelty of the study is revealing a correlation between severity of COVID-19 and the spike in the subclass of IgA – IgA2. The prevalence of this class in severely affected individuals was reported previously (and is rightfully so cited in the text) but the subclass data is novel. The weak point of the study is that it is just correlational data and no ex vivo studies on isolated neutrophils were performed. They would allow to directly show if indeed IgA2s induce NETs (and with what strength) when cells or Ab were isolated from patients with different forms of COVID-19. Although this Reviewer understands that the purpose of this paper was rather to report preliminary data on the correlation, in the light of missing direct measurements of NET release, placing a statement on their formation in the title seams to be rather a stretch. ecDNA can be used as a surrogate marker of NET formation if it is supported by other analyses, e.g. additional detection of NET proteins or preferably DNA-(NET)protein complexes. Itself, ecDNA might reflect on any cell death with necrotic morphology. For this I suggest to soften the title by either removing “NETs” or adding ecDNA, e.g. “…correlate with ecDNA/NET release…” to stay sound.

As pointed out by the authors, the patients with moderate and severe forms of the disease were significantly older. This was not to be avoided considering prevalence of these forms of COVID-19 in regard to age. I suggest however to collaborate on it more in detail. I believe, unlike IgGs, IgAs are in fact maintained in elderly or even increased in men (see for example DOI: 10.1258/000456303763046067 ) – could this further explain why older individuals produce more IgAs and this drives inflammation and then fatal outcome? Can you look back in your raw data and see any gender differences in IgA2s?

minor

-I would suggest to include also some brief discussion on IgMs

-p. 2 – the statement on neutrophils responding mostly to IgAs should be backed up with more references

Author Response

The topic of the manuscript is of high importance, it is still 2020. And it does contribute significant new data to the arising picture of the Sars-Cov-2 infection course. The novelty of the study is revealing a correlation between severity of COVID-19 and the spike in the subclass of IgA – IgA2. The prevalence of this class in severely affected individuals was reported previously (and is rightfully so cited in the text) but the subclass data is novel. The weak point of the study is that it is just correlational data and no ex vivo studies on isolated neutrophils were performed. They would allow to directly show if indeed IgA2s induce NETs (and with what strength) when cells or Ab were isolated from patients with different forms of COVID-19. Although this Reviewer understands that the purpose of this paper was rather to report preliminary data on the correlation, in the light of missing direct measurements of NET release, placing a statement on their formation in the title seams to be rather a stretch. ecDNA can be used as a surrogate marker of NET formation if it is supported by other analyses, e.g. additional detection of NET proteins or preferably DNA-(NET)protein complexes. Itself, ecDNA might reflect on any cell death with necrotic morphology. For this I suggest to soften the title by either removing “NETs” or adding ecDNA, e.g. “…correlate with ecDNA/NET release…” to stay sound.

We agree with the reviewer that it would be good to investigate the direct interactions of SARS-CoV-2 specific antibodies with neutrophils. However, the amount of plasma was very limited. We therefore have measured the activity of neutrophil elastase as a second marker for NET formation and compared it with the amounts of ecDNA (Figure S2). Both markers correlated with each other, indicating that ecDNA is a valid surrogate marker for NET formation.

As pointed out by the authors, the patients with moderate and severe forms of the disease were significantly older. This was not to be avoided considering prevalence of these forms of COVID-19 in regard to age. I suggest however to collaborate on it more in detail. I believe, unlike IgGs, IgAs are in fact maintained in elderly or even increased in men (see for example DOI: 10.1258/000456303763046067 ) – could this further explain why older individuals produce more IgAs and this drives inflammation and then fatal outcome? Can you look back in your raw data and see any gender differences in IgA2s?

According to the reviewer’s suggestion, we compared the levels of SARS-CoV-2 specific IgA1 and IgA2 in male and female patients with moderate or severe disease. It looks like IgA1 and IgA2 levels are slightly elevated in male patients compared to female patients, but the differences were not significant, probably due to the relatively small cohort size.

We did not see any correlation of SARS-CoV-2 specific IgA2 levels and age in patients with moderate disease. However, we found a significant increase in IgA1 and IgA2 levels with age in patients with severe disease. We have added these data to the manuscript as Figure S1. The increased levels of SARS-CoV-2 specific IgA1 and IgA2 in aged patients may reflect the association of older age with stronger disease severity that has been found in several studies.

minor

-I would suggest to include also some brief discussion on IgMs

We have included a small paragraph about IgM at the end of the discussion (page 7, line 239).

-p. 2 – the statement on neutrophils responding mostly to IgAs should be backed up with more references

We agree with the reviewer and included two additional references that show the superior response of neutrophils to IgA.

Round 2

Reviewer 1 Report

The authors have addressed all my concerns.